# Price Fluctuation, Protected Geographical Indications and Employment in the Spanish Small Ruminant Sector during the COVID-19 Crisis

**DOI:** 10.3390/ani10122221

**Published:** 2020-11-26

**Authors:** Irene Vidaurreta, Juan Orengo, Christian de la Fe, José María González, Ángel Gómez-Martín, Bernardino Benito

**Affiliations:** 1Department of Accounting and Finance, Faculty of Economics and Business, Regional, Campus of International Excellence “Campus Mare Nostrum”, University of Murcia, 30100 Murcia, Spain; irene.vidaurreta@um.es (I.V.); benitobl@um.es (B.B.); 2Department of Animal Production, Faculty of Veterinary Sciences, Regional, Campus of International Excellence “Campus Mare Nostrum”, University of Murcia, 30100 Murcia, Spain; jorengo@um.es; 3Ruminant Health Research Group, Department of Animal Health, Faculty of Veterinary Sciences, Regional, Campus of International Excellence “Campus Mare Nostrum”, University of Murcia, 30100 Murcia, Spain; angel.gomezmartin@uchceu.es; 4Animal Pathology Department, Instituto Agroalimentario de Aragón-IA2 (CITA), Faculty of Veterinary, Univeristy of Zaragoza, C/Miguel Servet 177, 50013 Zaragoza, Spain; jmgsovino@gmail.com; 5Gabinete Técnico Veterinario S.L. C/Isla conejera sn, 50013 Zaragoza, Spain; 6Microbiological Agents Associated with Animal Reproduction (ProVaginBio) Research Group, Department of Animal Health and Public Health, Faculty of Veterinary Sciences, University CEU Cardenal Herrera of Valencia, CEU Universities, 46113 Valencia, Spain

**Keywords:** COVID-19, small ruminants, economic impact, dairy, meat

## Abstract

**Simple Summary:**

This work assesses milk and meat price fluctuations from small ruminants during the coronavirus disease (COVID-19) crisis in Spain, using official data provided by the Interprofessional Dairy Organization (InLaC) and a representative sample of agricultural markets fixing prices per kilogram of lamb and goat kid meat. Data showed a drop in meat and goat milk prices during the period of confinement, which is only maintained for goat milk over the three following months. Similarly, analysis of data from the most important production areas, involving three protected geographical indications (PGI) of lamb meat, suggested that this quality label had a beneficial effect on sales and price stability in times of crisis. On the other hand, despite the impact of the COVID-19 pandemic on the Spanish labor market, the small ruminant sector did not seem to be affected in terms of the number of workers when comparing the period before and after the pandemic. All these factors are of interest for producers when making decisions regarding the management of flocks and adapting their marketing strategies for a down market or unexpected volatile circumstances.

**Abstract:**

Official milk prices in the Spanish small ruminant sector were used for 5 years (2015–2019) to analyze the effect caused by the coronavirus disease (COVID-19) crisis in 2020. Meat price fluctuations were also studied using the weekly prices officially provided by some of the main agrarian markets of the country (*n* = 6) in 2019 and 2020. Moreover, the sales and prices of three protected geographical indications (PGI) of lamb meat served to study the marketability when the products are sold or not under these quality labels in a crisis context. According to Spanish Government’s official communications, 2020 was divided in three periods of study (pre-COVID-19, total confinement and post-confinement). The evolution of employment in this subsector in 2020, as a direct consequence of this crisis, was also analyzed considering data provided by producers. Results showed an intra-annual seasonal effect for milk prices in 2020 for both livestock species, as observed in previous years. However, a negative economic impact on goat milk prices due to the pandemic was checked during the confinement and post-confinement months. Sheep milk prices remained stable. Lamb and goat kid meat prices showed a similar trend in comparison with 2019 during the pre-COVID-19 period. The total confinement period recorded a short interval of 1–2 weeks in which the prices declined, before the suspension of quotations in many markets. In contrast, once confinement was completed, meat prices for both ruminant species rapidly reached levels that existed before the coronavirus crisis. Overall data suggested the protective effect of the PGI marks on lamb meat. Lambs with a PGI had better 2020 prices than non-PGI lambs (+8%), regardless of the period analyzed. Moreover, with fewer lambs sold in 2020, there was a relevant drop in sales of non-PGI lambs vs. PGI (−19% vs. −2%) during the first 7 months. Finally, there was little or no readjustment of the workforce in the small ruminant flocks.

## 1. Introduction

The human pandemic COVID-19 was first identified in China at the end of 2019. Caused by the severe acute respiratory syndrome coronavirus (SARS-CoV-2), its contagiousness and the presence of asymptomatic carriers of the virus [1,2,3] epidemiologically explain the rapid spread of the infection throughout the world [4,5], as well as the differences observed in comparison with previous epidemics that occurred in this century caused by two viruses of the same genus (SARS-CoV-1 and Middle East Respiratory Syndrome -MERS-). Although potentially less pathogenic than other agents [6], SARS-CoV-2 forced many countries to adopt population-wide control measures, causing a social and economic effect which has not been sufficiently quantified to date.

Closure of borders, population confinement and interruption of non-essential economic activities were initially adopted by many countries. Later, this period of total confinement of the population has given way to the post-confinement, although restrictions on people’s movement and many economic activities continue [7], which has led to concern regarding the detrimental economic effect on very diverse sectors [8,9].

In general, livestock activities were considered essential to guarantee the food supply for the population in confinement. Consequently, the activity of small ruminants flocks was not interrupted. Moreover, ruminants are not apparently affected by SARS-CoV-2, as previously reported for MERS [10], although goats and sheep could be able to harbor and spread the virus [11]. Even without a direct effect on health, preliminary data obtained from a limited number of Spanish farms during the confinement showed that this sector suffered a short-term economic impact that resulted in lower milk and meat prices paid to producers, considering that the year 2020 started with high prices for small ruminant milk and meat, following the trend of high prices registered during the last years. [12]. Variation in prices is considered to be a normal aspect of well-functioning markets, but price volatility becomes problematic when price movements are large and unpredictable [13]. In this sense, the unexpected disruption of restauration and tourism caused by COVID-19 could partially explain this effect on prices reported by farmers [12]. However, interestingly, results also showed unpredictable differences of price for the same product depending on the geographical area studied in Spain, a fact previously reported in the literature [14]. In this sense, preliminary data provided by the farmers when the crisis started suggested that the presence of a protected designation of origin (PDO) or a protected geographical indication (PGI), quality labels linked to a geographical origin promoted by the European Union (EU) [15], may play a protective role in price maintenance in comparison with non-protected ones [12]. The EU quality policy aims at protecting the names of specific products to promote their characteristics linked to their geographical origin as well as traditional know-how. Differences between both products (PGI and PDO) are linked primarily to how much of the product’s raw materials must come from the area or how much of the production process has to take place within the specific region. This recognition could help producers to better market their products [15,16]. In this sense, recent data suggested that PGI may be an efficient tool to protect lamb farmers from fluctuating prices, thus assuring the fulfillment of positive externalities [17]. Moreover, the role of agrarian markets around the country, which set weekly meat prices and are also used as reference for commercial transactions [18], could also be the reason for price differences. All these factors should be analyzed in depth.

COVID-19 has dramatically affected employment in Spain. The Labour Force Survey (LFS) showed a sharp decline in employment in the second quarter of 2020, with an inter-annual decrease of 6.1% (equivalent to 1,198,000 employed), following the 1.1% increase observed in the first quarter of the year. In quarterly terms of the seasonally adjusted series, employment is estimated to have fallen by 6.7%, after the 0.4% fall in the first quarter of the year. In addition, during the second quarter, a very high percentage of workers were affected by temporary short-time work compensation schemes (TSTWCS), or they were self-employed in a situation of inactivity. Overall, workers in this situation increased to 3.4 million workers, representing 18.3% of total employment [19].

The small ruminant sector contributes in Spain only a 0.2% of the country’s gross domestic product (GDP) (2.7 for the primary sector as a whole), far from sectors such as the automotive industry (9%) or tourism (12%), which are non-essential activities. Specific data about the employment dynamic are scarce for this sub-sector. However, the global data on agrarian activities showed a continuous problem of generational replacement, reflected by a decrease in livestock units from 2005 to 2017 (58%). Employment decreased by 17% in the period 2000–2017, although registering better data than from 2015 to 2017. This decline has mainly occurred in family work, with a decrease in the total number of family units involved in agricultural work of 36% in the period 2000–2017 (an annual average of around 2%), while employed workers have increased by 12.5% in the same period [20]. According to the LFS data for the primary sector, at the end of the second quarter of 2020m the number of employed persons was 763,400, with employment below the level reached in 2019 and also with respect to the average for the period 2015–2019. Compared to the previous quarter, employment fell by 21,400 people (2.7%) [19]. When an economic crisis occurs, prices and employment are often affected [20]. In the context described, we have taken into account this economic precept to study whether the arrival of COVID-19 and its effect on small ruminant activities had any impact on employment.

Despite the above data, there are no in-depth studies on how this social and health crisis is affecting specific sectors that are qualified as essential to economic activity and, hence, provide the opportunity for our research. This work analyses the evolution of prices during the COVID-19 pandemic on small ruminant milk and meat markets in Spain during the first half of 2020, based on national and local official sources. Three periods have been defined (pre-COVID-19, total confinement and post-confinement), and the seasonality in prices and the correspondence between the sources of information were described and analyzed. The influence of marketability linked to PGI on lamb price fluctuation was also analyzed in detail as a factor of interest in maintaining prices during the health crisis. Thus, the data of sales and prices provided by three selected PGI lamb meat marketers facilitated the study of the differences with their non-PGI reference counterpart when the products are sold under these quality labels in a context of crisis. Finally, the effect of COVID-19 on employment in small ruminant flocks was examined.

## 2. Materials and Methods

### 2.1. Design

Milk and meat price fluctuations during the first 7 months of 2020 were analyzed using data provided by official sources of information in Spain (Spanish Interprofessional Dairy Organisation (InLac) for milk, and the main agrarian markets for meat). The pandemic COVID-19 reached Spain in 27 February 2020, population confinement was adopted by Spanish authorities on 14 March [21], and deconfinement was adopted on 28 April [22]. In this sense, three periods of analyses were established throughout the study: pre-COVID-19 (from 1 January to 13 March), total confinement (from 14 March to 28 April) and post-confinement (from 28 April to 31 July).

On the other hand, data provided by lamb meat marketers from the three of main geographical producing areas in Spain were evaluated: northeast (NE), southeast (SE) and southwest (SW). In each of these areas, data from lamb carcasses classified as non-PGI and PGI were collected. These data were analyzed to study the influence of these quality labels on meat sale and price fluctuation during the same periods above-mentioned.

Finally, the COVID-19 impact on the evolution of employment in small ruminant flocks was examined. The number of workers and number of animals per worker according to the type of flock during the pre-and post-COVID-19 periods were analyzed using data provided by producers.

### 2.2. Milk Prices Published by InLAC

Monthly prices for goat and sheep milk (2015–2020) were provided by InLac (the Spanish Interprofessional Dairy Organization (Interprofesional Lechera de España)), recognized by the Ministry of Agriculture, Fisheries and Food. InLac groups the Spanish dairy sector, whose members represent all the production branch, producers, processors, cooperatives and industries [23]. Briefly, data by InLac were used to describe the intra- and inter-annual price variation of the previous 5 years, as well as the price fluctuation in 2020.

### 2.3. Evolution of Meat Prices on Agrarian Markets

The sale prices of lamb and kid meat obtained at the first auction were analyzed on the basis of weekly prices established for a representative sample of the main agrarian markets in the country (*n* = 6): Talavera de la Reina, Albacete, Segovia, Burgos, Villalpando and Murcia. Agrarian markets were selected for its importance and the availability of information. In this way, for better representativeness within each product, at least 3 geographically distant markets were considered. They provide guidance to boost market prices, as well as to make it easier for buyers and/or marketers to carry out their purchasing operations. In certain sectors, products or areas, they represent the only or main point of discussion, information or contact between operators [24]. Data analyzed were the weekly prices per kilogram of lamb or goat kid produced from January 2019 to July 2020. Taking into account the meat products consumed in Spain, three products were used as reference: 2 for lambs (10–12 kg, and 19–25 kg of live weight (LW)) and 1 for goat kids (7–9 kg LW).

### 2.4. Influence of PGI on Lamb Sale and Price Fluctuation

According to the geographical area, three PGI sources were chosen: Ternasco de Aragón (NE), Cordero de Extremadura (SW) and Cordero Segureño (SE). These areas were selected to analyze in depth the differences in sale and price fluctuation between PGI and non-PGI lambs. Data were analyzed inter- and intra-annually in an aggregated way, considering lamb carcasses with the same characteristics of weight. Thus, the average lamb price was calculated by weighing individual prices in relation to the number of lamb carcasses of each area. The total number of lamb carcasses analyzed (from 10.1 to 13 kg CW) represented 8.5–10% of the Spanish market according to data from January to July 2019 [25].

These PGIs were selected for several criteria: (1) its interest and economic weight; (2) the type of product—only one type of lamb, similar and comparable between the PGI and its non-PGI counterpart, was produced under the rules of each Regulatory Council; and (3) the information of the PGI vs. non-PGI was available. The PGI “Cordero de Extremadura” came from the breed Merino under extensive and semi-extensive production systems, where light carcasses are produced not exceeding 14 kg for females and 16 kg for males [26]. The PGI “Cordero Segureño” came from the breed Segureña in a semi-extensive production system, in which light carcasses from 9 to 13 kg are produced [27]. Finally, the PGI “Ternasco de Aragón” came from the breeds Ojinegra de Teruel, Roya Bilbilitana, Rasa Aragonesa, Maellana and Ansotana. Light carcasses from 8 to 12.5 kg are raised in a semi-extensive production system [28].

### 2.5. Evolution of Employment on Flocks

The impact of this pandemic event on farms’ employment was examined by analyzing the number of workers employed in the pre-and post-COVID-19 months. This information was provided by different farms monitored with technical–economic management software. These data were analyzed in (1) absolute and (2) relative terms. This latter expressed as a function of number of animals per worker.

### 2.6. Statistical Analysis

Data were analyzed using the SPSS program (SPSS Inc., Chicago, IL, USA). Price fluctuation data were graphically represented and described within each period of analysis: pre-and post-COVID-19. This latter was, in turn, differentiated and subdivided into total and post-confinement period (with strict or partial confinement measures, respectively). Employment data were summarized by the main descriptive statistics, and then compared by period using a test for paired samples (“within-flocks” observations). A preliminary analysis showed that employment data were not normally distributed, thus, two non-parametric tests were carried out to determine whether the median of pairwise differences between periods was different from zero: the Wilcoxon signed-rank test [29] and a trinomial test when there were many ties or zeros in the paired data [30]. The trinomial test was performed by using Microsoft Excel (Microsoft, Redmond, WA, USA) with the Real Statistics Resource Pack Software Release 6.8 (available at http://www.realstatistics.com/) [31]. The significance level was set at *p* < 0.05.

## 3. Results

### 3.1. Analysis of Spanish Small Ruminant Milk Prices from 2015 to 2020

Prices officially recorded for goat and sheep milk by InLac from 2015 to 2020 are shown in Figure 1. A seasonal trend was found with prices decreasing from January to March–April, followed by a period of stable prices for 3–5 months, and, finally, there was a continuous and constant increase in milk prices until the end of the year (with a peak in November). This intra-annual seasonal effect was similar for both livestock species, but it is necessary to remark that (1) although there were significant inter-annual price fluctuations, average prices of milk were higher for sheep; (2) the seasonal effect seemed to be more pronounced for goat milk prices. Analyzing the milk market in 2020 (Figure 1), prices showed the same seasonal trend previously described. The highest goat milk price for the first months of the year was recorded in early 2020 (January–February). However, while sheep milk prices were higher than in 2019 during the first half of the year, goat milk prices fell in April 2020 compared to 2019 values. Moreover, this trend of lower prices for goat milk was also maintained over the following months, corresponding to the post-confinement period.

### 3.2. Analysis of Meat Prices Provided by Agrarian Markets

A seasonal trend was found in lamb and goat kid meat prices in 2019. Prices decreased at the beginning of the year for both species, then a stationary level was observed from April to June, and finally there was a continuous price increase from October to December, when the highest annual prices were recorded. In 2020, the lamb and goat kid meat prices showed a similar behavior in comparison with 2019 during the pre-COVID-19 period. However, the total confinement period recorded a short interval of 1–2 weeks in which the prices declined, before the suspension of quotations in many markets. In contrast, the post-confinement period was characterized by a price recovery that in a few weeks reached and even exceeded last year’s prices. This trend in prices was observed in all the agrarian markets analyzed.

Prices recorded during the study were also different according to the geographical area and type of product. Sometimes, these differences were up to 1 EUR/kg for the same weight and week depending on the agrarian markets (Figure 2A). In this sense, taking into account the three types of product analyzed, the weekly prices of 10–12 kg lambs tended to fluctuate much more than lambs from 19 to 25 kg live weight (Figure 2A,B); price differences among markets were suggested to be higher for lighter lambs. The price of goat kid meat followed a similar trend to that of light lamb, although the price fluctuation seemed smaller (Figure 2C).

### 3.3. Differences in Sales and Prices among PGI and Non-PGI Lambs

The absolute number of lambs sold during the period of study was 13% lower in 2020 than in 2019, with a more marked fall in the non-PGI vs. PGI lambs (−19% vs. −2%) (Figure 3). Analyzing each period, the pre-COVID-19 sales in 2020 (1 January–13 March) decreased compared to the same period in 2019 (18%). However, sales of non-PGI lambs fell by 34% compared to PGI lambs, which showed an increase of 18% (Figure 3). During the confinement period (14 March–28 April), the fall reached 21%, focused again on the non-PGI lambs (−33% vs. −3% for PGI lambs). Finally, the post-confinement period showed a drop in sales of 4%, partially compensated by higher sales of non-PGI lambs (29 April–31 July).

On average, lamb prices were 6% higher in 2020 vs. 2019 for the whole period (Figure 4). Moreover, PGI lambs showed higher prices than non-PGI lambs (+8%) in 2020, regardless of the period analyzed (Figure 4A). Pre-COVID-19 lamb prices in 2020 were suggested to be higher than in 2019 for both non-PGI and PGI lambs (+21% and +10%, respectively). Nevertheless, during the first months of 2020, higher prices were recorded for PGI lambs than those obtained for non-PGI lambs (+10%). This difference remained during the confinement period (+8%), although both of them showed similar falls in prices (−11%) in comparison with the pre-COVID-19 period, reaching 2019 prices. Finally, prices recovered during the post-confinement period (+14% in 2020 vs. 2019), where PGI vs. non-PGI lambs showed again the best price (+7%) (Figure 4A).

A relative analysis of price fluctuation in each type of lamb regarding the initial prices of every year (Figure 4B) showed a fall for both groups during the confinement, but tending to be more pronounced for non-PGI lambs (2020 vs. 2019 prices: −17% and −11% for non-PGI and PGI lambs, respectively). The post-confinement period suggested a recovery in price for PGI lambs (+1%), while it pointed to slight losses for non-PGI lambs (−3%). Overall, results showed that the price of PGI lambs seemed to be more stable and always higher in comparison with non-PGI lambs.

### 3.4. Analysis of Employment on Flocks

The analysis of the number of employees present on the farms of our study showed that 90% of them (52 out of 58 flocks sampled) did not suffer changes in their staff in the post-COVID-19 months (total confinement and post-confinement) in relation to the pre-COVID-19 period (Table 1). On the other hand, only 5% of flocks showed an increase in the number of workers (corresponding to three dairy sheep farms), while there was a reduction of the same percentage of staff in caprine flocks.

On average, the number of workers and animals per worker in small ruminant flocks was around 4 and 240, respectively, regardless of the period analyzed. Moreover, the number of workers and their rank was numerically lower in meat vs. dairy production (Table 1).

The statistical comparison of the medians did not show significant differences for both variables, number of workers and number of animals per worker (Table 2). The non-significant results from the Wilcoxon sign rank test were supported and confirmed by a trinomial test, which took into account that there were many ties or zeros in the pairwise differences (*p* ≥ 0.05).

## 4. Discussion

The analysis of official prices for the first 7 months of 2020 confirmed that the crisis caused by the coronavirus had a relevant economic impact on the small ruminant sector. These data also showed that the impact was not only limited to the confinement period, but also it was maintained during the post-confinement months in some sub-sectors. Indeed, prices in the post-confinement period showed that, while the effect was constant until July in dairy goat, meat market prices recovered rapidly to pre-coronavirus crisis levels for both ruminant species. Data collected also showed the seasonality of milk and meat prices [32]. In this line, although the pandemic reached Spain in a phase of seasonal decline in prices, the drop was greater in comparison with the previous year.

Variation in prices is considered to be a normal aspect of well-functioning markets [13]. In fact, not all price variations are problematic, and agricultural prices vary because production and consumption are changing [33]. In addition, other factors such as fuel price volatility or policy decisions may also lead to higher price fluctuations [13,33,34]. In economic theory, price volatility involves two principal components: variability and uncertainly, the former describing overall movement and the latter referring to movement that is unpredictable [35]. It is a way of measuring the variability of the price or quantity of a product over time [13]. The results obtained suggested that the extreme measures adopted to control the pandemic COVID-19 caused an unpredictable shock of consumption of some products that may have been transmit into price variability [33], although further in-depth economic analysis of all the possible factors associated should be conducted. These data highlight the need to develop measures and policy responses to help producers fight price volatility [13,33,34,36,37]. In this regard, the efficiency degree of the markets is also important in order to adopt stabilizing measures that can dissipate any shocks with no degree of persistency of volatility [17].

Milk price fluctuation analyses, using data provided by the Spanish Interprofessional Dairy Organisation [23], showed differences between both small ruminant species during 2020, and not only during the strict confinement period in March–April, as previously pointed out by some producers [12]. These differences persisted from May to July for dairy goats. In contrast, sheep milk price remained stable or even increased in 2020 vs. 2019. This negative effect on goat milk prices during confinement, attributed to inexistent consumption of livestock products linked to restauration and tourism, could not be enough to explain why the prices did not rise when economic activities returned. Interestingly, 2020 prices of this product showed that, although they fell by around 15 cts EUR per liter in April vs. February, they were still higher than the average price for the period 2015–2019, thus opening the door to a price adjustment in this sub-sector. In fact, the year 2020 began with the highest historical prices in the dairy goat sector.

Lamb and goat kid meat prices recovered in May–June after the period of strict confinement in Spain. In some cases, they even overpassed 2019 prices. These data seem to rule out the hypothesis of a permanent effect on the small ruminant meat sub-sector following the sharp fall in sales and activities in March–April 2020 [12]. This fast recovery could be supported to several factors: (1) the gradual increase in domestic consumption [38]; (2) the opening of new distribution channels such as “close contacts” or B2C models to sell and distribute products directly to the consumer, as well as the e-commerce boom triggered by COVID-19 [38,39,40]; and/or (3) the partial recovery of tourism, mainly domestic, during the post-confinement phase [41]. The analysis of meat prices provided by several agricultural markets also showed differences among them, even for the same product in nearby geographical areas. In this sense, differences among regions and products in the speed and degree of price movements in regional or local markets were previously reported, thus suggesting that both price levels and degrees of variability may differ from place to place [41]. In our case, the weekly price fluctuation seemed to also be important inside the same market, especially for some products such as the lighter lamb (10–12 kg LW). Differences linked to destination of these lambs (slaughterhouses vs. feedlots), or the quality of the lamb sold in certain areas (depending on the breed of origin) [42], may explain these variations. Price fluctuations tended to be minor for goat kids and heavier lambs. Heavier lambs can be sold at different weights, and commercial outlets linked to international markets can influence their versatility and price stability. On the other hand, goat kids are commonly produced and sold with similar age and weight all over the country.

The pre-COVID-19 period was exceptional for lamb prices, although they fell in March–April. This drop in prices occurred despite the coincidence of two religious festivals that usually cause an increase in demand (Easter and the beginning of Ramadan). In this context, results suggested that selling lamb meat products under the protection of a PGI had a beneficial effect on maintaining prices and sales during the period of crisis studied. This finding is in line with the much weaker price volatility recently reported for the lamb Cordero de Navarra in the PGI vs. non-PGI system for the period 2011 to 2018 [17]. Furthermore, these data are in agreement with previous results showing that the PGI product is better able to stand up to consumer confidence crises [43]. In this sense, a designation of origin (DO) linked to milk or cheese had a positive influence on prices paid to producers, as reported in Manchego sheep flocks [12]. Conversely, this protective effect during a health crisis has been scarcely studied for meat products. Regardless of the geographical area and particularities of each PGI, the lamb meat sold under the protection of a PGI pointed to better prices for all the periods studied and cushioned its fall during the total confinement. In addition, sales suggested better relative results during the total confinement for PGI lambs, where sale reductions were 3% compared to 33% for non-PGI lambs. This positive effect was probably due to the strong promotional campaigns undertaken by the three Regulatory Councils, focused on both direct and local sales (Extremadura and Segureño lambs) and traditional shops and supermarkets (Ternasco de Aragón and Extremadura lambs), or even the consumer trust in the added value of these products [16,19,43]. This fact is valuable because promoting PGI lamb could protect the farmer from fluctuating prices [19]. Therefore, competitive advantages of each PGI linked to the greater consumer knowledge should be considered by producers when developing and managing their flocks with one or another breed, as a form to obtain added-value products [17,29,44,45]. Despite this, in terms of the number of lambs sacrificed in 2018 in Spain, PGI vs. non-PGI lamb accounted for 7.3% [26,46]. Regarding this, some authors suggested that the public information available describing the characteristics of geographical indication products is still scarce [37].

On the other hand, this research suggests that the readjustment of the workforce in sheep and goat farms was minimal or non-existent, despite the negative economic effect observed for the first months of the pandemic. A similar situation was observed following the financial crisis that began at the end of 2007, in the sense that the employment trend in the primary sector in those years was not influenced by the negative effects of that crisis [14]. Specific employment data for the small ruminant sector are scarce in the literature, although it is well-known that agrarian activities suffer from a persistent problem of generational replacement [20]. Despite this, it is remarkable that while employment linked to other sectors (e.g., tourism and the automotive industry) or even other areas of the primary sector was affected by the crisis, layoffs in the small ruminant sector were minimal according to the data analyzed. This fact should be considered in order to promote employment in this sector, where it is sometimes considerably complex to find qualified workers. In this respect, according to the primary sector data from Spanish LFS [19], it should be mentioned that agriculture and livestock employed 763,400 people at the end of the second quarter of 2020, having decreased by 21,400 people (2.7%) compared to the first quarter of this year and by 44,000 (5.9%) compared to 2019.

## 5. Conclusions

Official data showed a negative effect of the pandemic on the goat milk market, where there was no sign of recovery, while the meat market recovered rapidly after the end of strict confinement. Moreover, current results also suggested that the production of lamb meat under the protection of a PGI had a positive effect on maintaining prices and sales. The impact of the crisis on employment on farms was non-significant, despite the negative effects reported.

## Figures and Tables

**Figure 1 animals-10-02221-f001:**
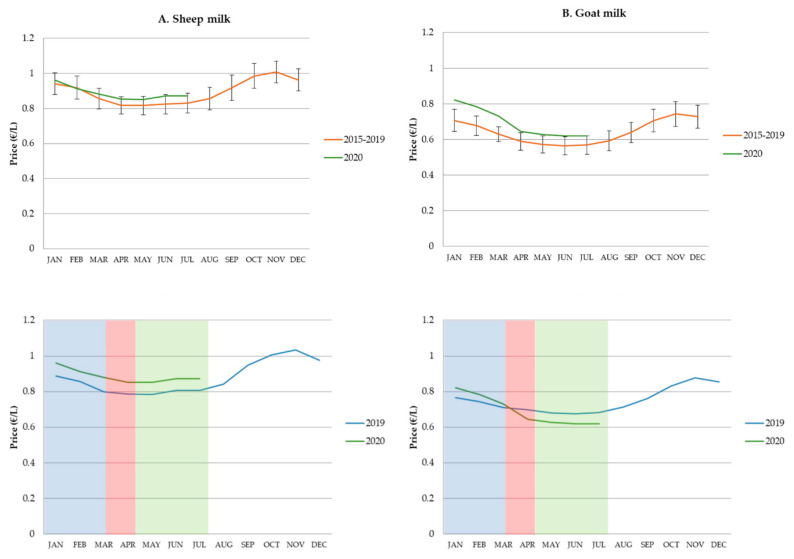
Average monthly price per liter (EUR/L) of sheep (**A**) and goat milk (**B**) reported by Spanish Interprofessional Dairy Organization from January to December 2015–2019 (error indicating the 95% confidence interval for the monthly means), and the inter-annual comparison of 2020 vs. 2019. Blue, red and green shaded area corresponds to the period from 1 January to 13 March (pre-COVID-19), from 14 March to 28 April (total confinement), and from 29 April to 31 July (post-confinement), respectively.

**Figure 2 animals-10-02221-f002:**
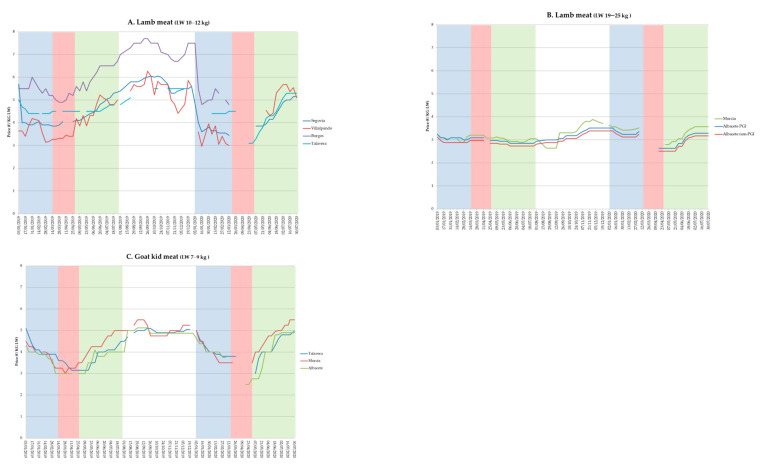
Weekly price per kilogram of live weight (EUR/kg) for 10–12 kg (**A**) and 19–25 kg lamb (**B**), and 7–9 kg goat kid (**C**), reported by main Spanish agrarian markets in 2019–2020. Blue, red and green shaded area corresponds to the period from 1 January to 13 March (pre-COVID-19), from 14 March to 28 April (total confinement), and from 29 April to 31 July (post-confinement), respectively.

**Figure 3 animals-10-02221-f003:**
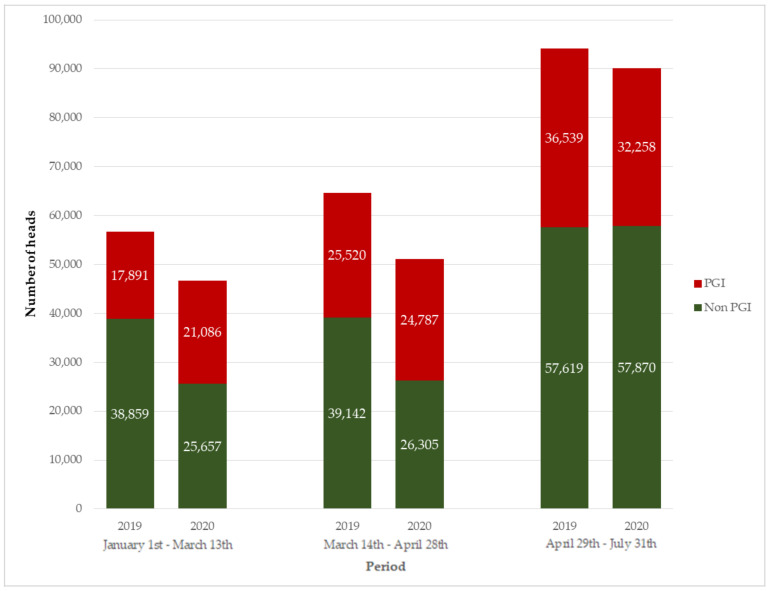
Evolution of total sales (number of heads) by non-PGI and PGI lambs in 2019–2020 in Spain (PGI: protected geographical indication). Source: Regulating Council of Ternasco de Aragón, Cordero de Extremadura and Cordero Segureño.

**Figure 4 animals-10-02221-f004:**
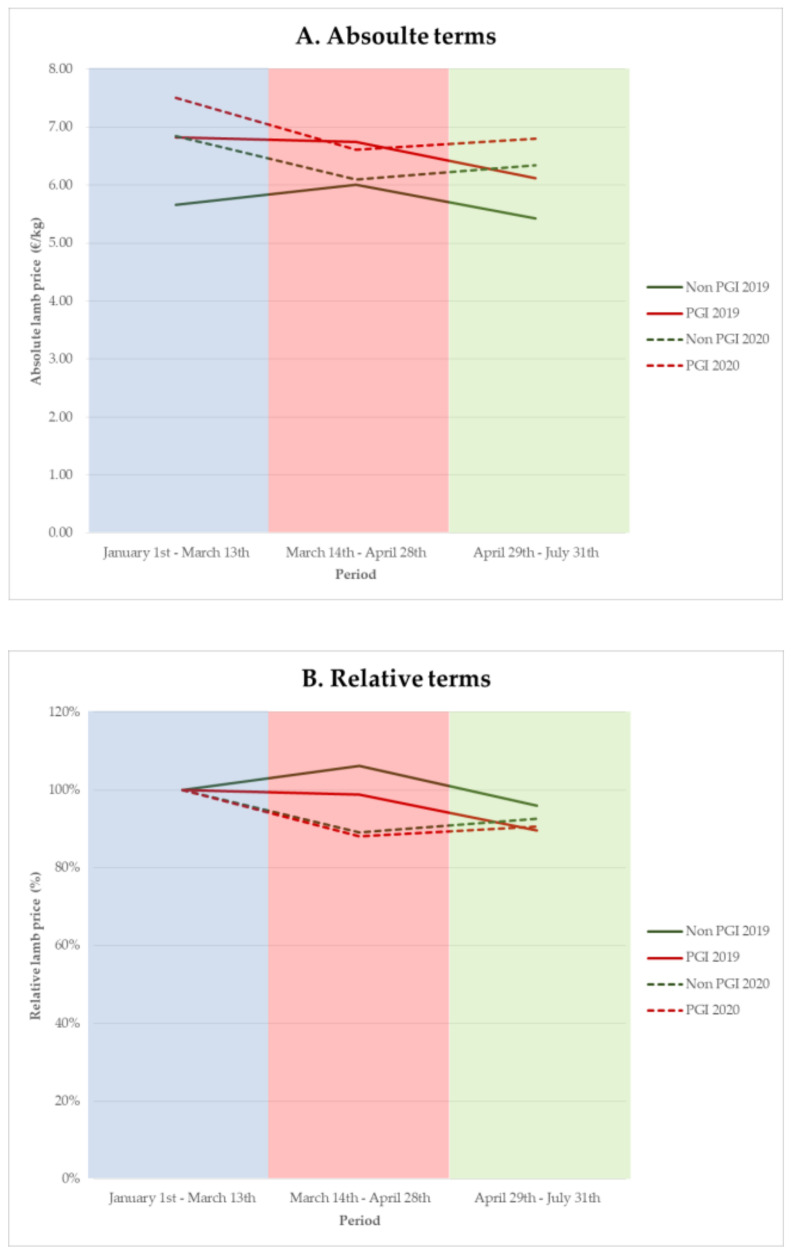
Evolution of average price by non-PGI and PGI lambs in 2019–2020 in Spain (PGI: protected geographical indication), in absolute (EUR/kg) (**A**) and relative (%) (**B**) terms. Relative prices are expressed as a percentage in relation to the initial period prices of every year. Blue, red and green shaded area corresponds to the period from 1 January to 13 March (pre-COVID-19), from 14 March to 28 April (total confinement), and from 29 April to 31 July (post-confinement), respectively.

**Table 1 animals-10-02221-t001:** Descriptive statistics (mean, standard deviation (SD), minimum (Min). and maximum (Max)) for the number of workers and number of animals per worker according to the type of flock during the pre-and post-COVID-19 period.

Production System and Species	Flocks (*n*)	No. of Workers								No. of Animals per Worker							
		Pre-COVID				Post-COVID				Pre-COVID				Post-COVID			
		Mean	SD	Min	Max	Mean	SD	Min	Max	Mean	SD	Min	Max	Mean	SD	Min	Max
Dairy sheep	22	5.0	1.4	3	8	5.2	1.6	3	8	248	97	105	520	244	102	79	520
Meat sheep	13	2.2	0.8	1	3	2.2	0.8	1	3	327	142	105	604	327	142	105	604
Dairy goat	23	3.8	3.0	1	13	3.6	2.9	1	13	177	70	40	300	188	67	40	300
Total	58	3.9	2.4	1	13	3.9	2.4	1	13	238	115	40	604	240	114	40	604

**Table 2 animals-10-02221-t002:** Median of the number of workers and number of animals per worker according to the type of flock during the pre-and post-COVID-19 period, with inter-period comparisons.

Specie	Flocks (*n*)	No. of Workers (A)					No. of Animals per Worker (B)					*p*-Values ^2^	
		Median		Pairwise Differences ^1^			Median		Pairwise Differences ^1^				
		Pre-COVID	Post-COVID	Positive	Ties	Negative	Pre-COVID	Post-COVID	Positive	Ties	Negative	A	B
Sheep	35	4	4	3	32	0	260	260	0	32	3	0.102	0.109
Goat	23	3	3	0	20	3	170	178	3	20	0	0.109	0.109

^1^ Number of paired differences (or “within-flocks” observations) ranked as positive or negative, including ties. ^2^ Wilcoxon signed-rank test comparing medians for paired samples.

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
