# Peer review of "Price Fluctuation, Protected Geographical Indications and Employment in the Spanish Small Ruminant Sector during the COVID-19 Crisis"

_animals, 2020, doi:10.3390/ani10122221_

Round 1
Reviewer 1 Report
Overall, a timely and solid piece. I appreciate that the authors used data that was available and didn't over analyze the situation. Very readable. I have put minor edit in the pdf (attached).
General comments: be consistent with COVID-19 in the text. Sometimes it was COVID or COVID-19. Pick one and roll with it.
Be consistent with formatting your figures. There are issues with boarders, labels, and font. I put the edits in the pdf
Be consistent with formatting your references, i circled ones on the pdf that stuck out.
Minor comments: I edited some wording/dictation stuff, but there a few instances where clarity is needed and i noted that.

Author Response
"Please see the attachment."

Reviewer 2 Report
Summary
This paper presents a descriptive statistical analysis on the prices of milk and meat products from small ruminants in the dawn of the COVID-19 crisis in Spain.
General comments:
Overall, I find the topic of the paper interesting. However, the paper in its current form suffers from serious issues that could undermine its contribution. For that reason, I urge the authors to rethink their research project based on the comments detailed below which, hopefully, could improve the quality of their contribution.
I recommend a thorough revision of the introduction of the paper. I would expect authors expand more on the relation between prices and employment. Is their approach economic theory-based? I also think more on the past context of prices for the sector would be more than interesting for the reader. Authors shall revise the part related to European food quality labels to better explain their original purpose and perhaps offer a brief description of those, which are going to be analyzed. Further, showing production and sales representativeness in Spain with respect to the non-PGI would be advisable to show the reader to what extent quality is important in terms of trade and production in Spain. Regarding the employment in Spain, authors should revise also that part. Are there any recent papers (before COVID) dealing with the objective they attempt to analyze here? The literature review offered by authors is scarce and it seems hardly fair to let the limited literature provided take all the weight of the basis of the paper. Authors shall conduct a more detailed literature search looking for studies dealing with volatility of prices and/or food products differentiated with a quality label.
Regarding Section 2.3, what prices are you dealing with? Farm prices, first auction sale prices, wholesaler prices or consumer prices? Why did you chose those agrarian markets? Are they the most important in Spain or you picked them because you got access to their data? Prices per kg are liveweight or carcass weight? (Though it is reported in the respective figures, so I recommend homogenize).
Regarding subsection 2.4, why did you choose those PGI lamb breeds? What about the PGI Castilla y Leon lamb? Your choice is based on the fact that they are the most commercialized or most produced in the country? Authors should revise the reported % of lamb carcasses representativeness with the stated reference, as it seem not to be correct.
As for the use of the Wilcoxon signed-rank test. It is true that it is a useful nonparametric tool but I was not able to find anywhere neither the null hypothesis nor the hypothesized value for comparison used. Authors could expand on their methodological approach (section 2.6). Where did you performed the test? In SPSS, R, Stata? Did you check whether your data meet all the assumptions that allow you to use the test? Do the distribution of the differences between your groups are symmetrical in shape?
Regarding Results section: The analysis of the impact of the virus is merely based on visual inspection of figures and comparison of figures in tables reporting basic descriptive statistics. A nonparametric test is used but no null hypothesis has apparently been defined.
Furthermore, in subsection 3.3, regarding the analysis of the Pre-COVID, authors seem to point out that the difference between PGI and non-PGI is only based on the quality label just by visual inspection. I believe that this allegation is quite strong as there are no theory-based arguments, econometric modeling, hypothesis testing or literature references supporting it.
In relation to the Discussion of the results. Since the analysis conducted in the paper is a mere descriptive exercise, the discussion of the results should be restricted to its nature and choose carefully the terms accordingly without pretending to be too ambitious. Moreover, some allegations/statements should be reconsidered, specifically those referred to the causality of changes as they are not supported with proper economic theory or econometric modeling or hypothesis testing. There seems not to be the adequate approach used to properly identify causality of the changes. For instance, it seems hardly fair to let [24] take all the weight of the statement. Do you find any references providing us further insights (related to the meat sector or to PGI-PDO-TSG or at least organic)?
Minor comments
Paragraph “As regards […] (2.7%) [15]” gives official figures that are different from those reported in the last sentence of the last paragraph in section 4. “In this respect, […]”
Authors should define p (the p-value) and the significance is set at 5% nominal level.
What is the data source of Figure 3?
Given the fact that there is no widely definition of volatility in the related literature (see, among others, Serra and Zilberman, 2013), authors should define the term or consider using variability instead.
How can you be so sure that the effect is permanent without proper modelling and hypothesis testing?
Number formatting in figures is not in English system. Authors should redo.
References:
Serra, T., and Zilberman, D. (2013). Biofuel-related price transmission literature: A review. Energy Economics 37: 141–151. doi:10.1016/j.eneco.2013.02.014
Author Response
"Please see the attachment."
Round 2
Reviewer 2 Report
The paper has been adequately edited and I find that all my previous concerns in your original paper are corrected.